# Deep mutational scanning identifies Cas1 and Cas2 variants that enhance type II-A CRISPR-Cas spacer acquisition

Raphael Hofmann [1] ✉, Calvin Herman[1], Charlie Y. Mo [1,3], Jacob Mathai[1] & Luciano A. Marraffini [1,2] ✉

A remarkable feature of CRISPR-Cas systems is their ability to acquire short sequences from invading viruses to create a molecular record of infection. These sequences, called spacers, are inserted into the CRISPR locus and mediate sequence-specific immunity in prokaryotes. In type II-A CRISPR systems, Cas1, Cas2 and Csn2 form a supercomplex with Cas9 to integrate viral sequences. While the structure of the integrase complex has been described, a detailed functional analysis of the spacer acquisition machinery is lacking. We developed a genetic system that combines deep mutational scanning (DMS) of *Streptococcus pyogenes cas* genes with a method to select bacteria that acquire new spacers. Here, we show that this procedure reveals key interactions at the Cas1-Cas2 interface critical for spacer integration, identifies Cas variants with enhanced spacer acquisition and immunity against phage infection, and provides insights into the molecular determinants of spacer acquisition, offering a platform to improve CRISPR-Cas-based applications.

Many prokaryotes harbor CRISPR-Cas systems, which consist of clustered, regularly interspaced, short palindromic repeat (CRISPR) loci and a flanking operon of CRISPR-associated (*cas*) genes[1]. CRISPR-Cas systems provide acquired immunity to protect the host against invading bacteriophages (phages) and plasmids[2,3]. The CRISPR locus contains short sequences of foreign DNA, called spacers, located between repeat sequences, that are transcribed and processed into CRISPR RNAs (crRNAs)[4]. CrRNAs associate with Cas nucleases and guide them to complementary sequences within invading nucleic acids[5,6] known as protospacers. Effective targeting in some systems also requires the presence of a protospacer-adjacent motif (PAM)[2].

New spacers are extracted from prespacer sequences in the invading DNA and inserted into the CRISPR array by the Cas1-Cas2 integrase complex[7,8], a process that generates a genetically stable memory of the infection, with the CRISPR array serving as an immunological repository[2]. Spacer integration occurs in a polarized manner, with new sequences being added upstream of the first repeat of the array, where the CRISPR leader sequence is located[9,10]. CRISPR-Cas systems can be classified into seven types based on characteristic *cas*

genes[11,12]. While Cas1 and Cas2 are conserved across many types of CRISPR-Cas types, additional Cas proteins and host factors are required for spacer capture, processing and integration[13]. Spacer acquisition has been studied primarily in type I-E and type II-A CRISPR-Cas systems. In the type I-E system, selection of functional spacers is directly mediated by the Cas1-Cas2 integrase, with Cas1 conferring PAM specificity to facilitate this process[14]. In other type I systems (types I-A to I-D), the integrase complex also includes the Cas4 protein, which ensures the recognition of PAM-containing prespacers[15,16]. In addition to the Cas1-Cas2 integrase, type II-A loci encode the crRNA-guided nuclease Cas9, which has been widely adopted for genetic engineering applications[17], and Csn2 (ref. 13). Type II-A loci also produce a trans-activating CRISPR RNA (tracrRNA) that functions as a Cas9 co-factor required for DNA cleavage and crRNA maturation[18]. The tracrRNA contains a sequence complementary to the CRISPR repeat that enables the formation of a dsRNA with the repeats present in the transcript of the CRISPR array, known as the CRISPR RNA precursor, pre-crRNA. RNase III cleaves the dsRNA regions of pre-crRNA at the repeat sequences, liberating mature crRNAs that remain associated

[1]Laboratory of Bacteriology, The Rockefeller University, New York, NY, USA. [2]Howard Hughes Medical Institute, The Rockefeller University, New York, NY, USA. [3]Present address: Department of Bacteriology, University of Wisconsin, Madison, WI, USA. ✉e-mail: rhofmann@rockefeller.edu; marraffini@rockefeller.edu

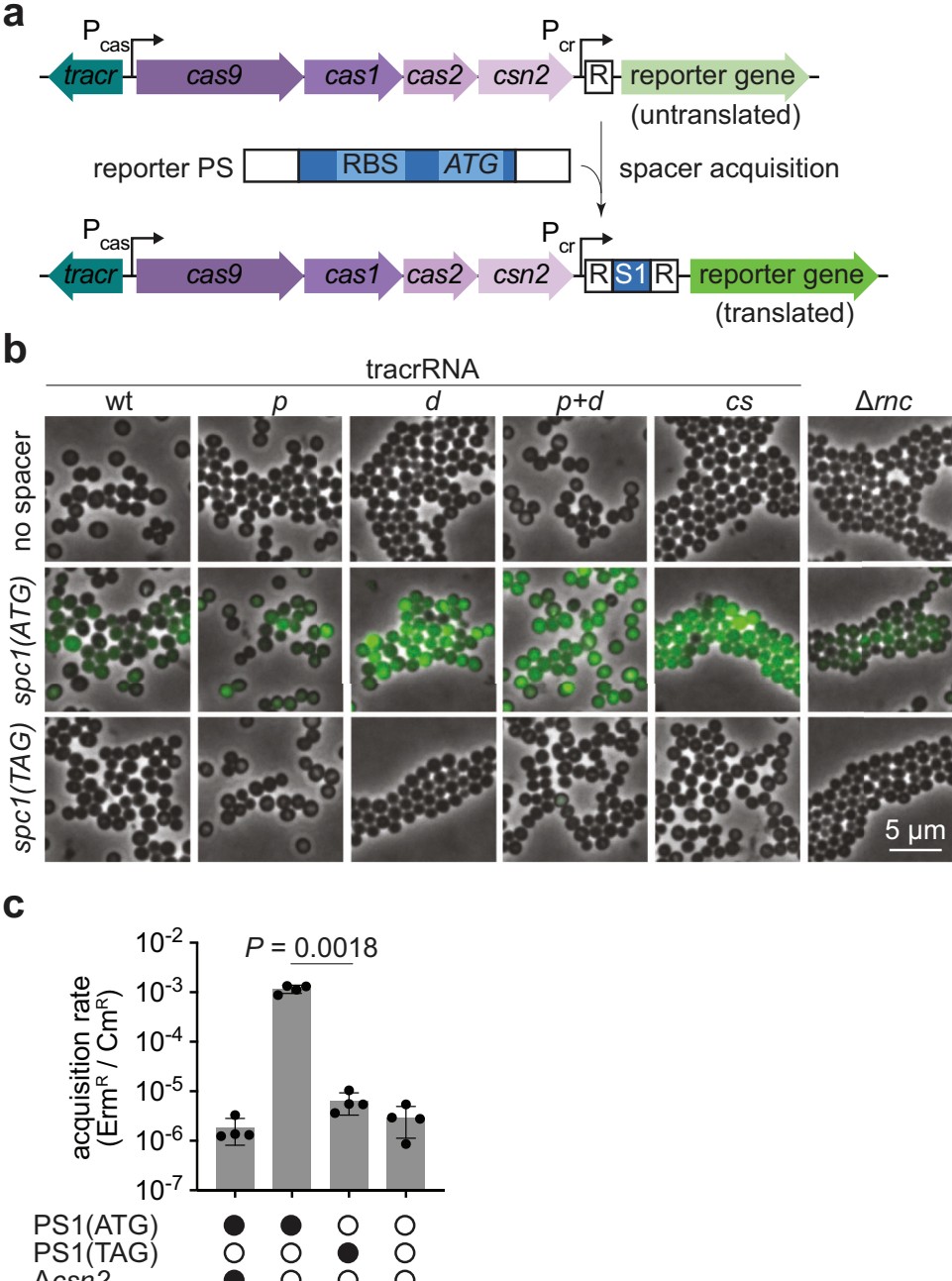

**Fig. 1 | Reporter strain for type II-A CRISPR-Cas spacer acquisition. a** Schematic representation of the reporter strain. Spacer acquisition from a prespacer (PS) containing a ribosome binding site (RBS) and a start codon (ATG) results in the installation of a translation start site, leading to downstream reporter expression. $P_{cr}$, CRISPR array promoter; $P_{cas}$, *cas* operon promoter; R, repeat. **b** Fluorescence microscopy of *S. aureus* cells carrying a reporter plasmid expressing mNG (green fluorescence) with a single CRISPR repeat, a spacer with a translation start site (*spc1(ATG)*) or a spacer with a defective start site in which the ATG start codon was replaced by a stop codon (*spc1(TAG)*). Each spacer construct was combined with different tracrRNAs or transformed into staphylococci lacking RNase III (Δ*rnc*). Representative images are shown (*n* = 2 biological replicates). **c** Spacer acquisition rate as determined by the ratio of CFUs on erythromycin and chloramphenicol ($Erm^R/Cm^R$) after electroporation of reporter strains with the indicated prespacers (or no prespacer) in the presence or absence of *csn2*. Individual data points are shown with error bars representing the mean ± s.d. of four technical replicates.

with the tracrRNA and thus with Cas9 (ref. 18). The tracrRNA is transcribed from two different promoters, generating a short and long form[19]. The long form contains a region complementary to the promoter of the *cas* genes (Pcas, Fig. 1a) that enables Cas9 to function as a tracrRNA-guided transcription repressor of the type II-A operon. Prespacer capture in type II-A systems is carried out by a Cas1-Cas2-Csn2-Cas9-tracrRNA integrase supercomplex[20]. Overexpression of this supercomplex, either through the replacement of Pcas by a strong promoter[21,22], or through the deletion of the promoter that controls transcription of the long tracrRNA form[19], enhances spacer acquisition. Spacer acquisition is also stimulated by free dsDNA ends and DNA breaks[23], which can be generated by restriction endonucleases[24], RecBCD- or AddAB-mediated processing of injected phage DNA[25] and stalled replication forks[26], and by Cas nucleases themselves[27]. The type II-A supercomplex has been studied biochemically and structurally. It has been demonstrated that Cas1 possesses the integrase activity[28,29], Cas9 (loaded with the tracrRNA) is key to identify and mark functional prespacers with the correct PAM[28,30], Csn2 forms a tetrameric ring[31,32]

that binds free DNA ends[33] and interacts with Cas1 and Cas9 (refs. 34–36), and Cas2 plays a structural role. While the roles of the components of the integrase supercomplex have been defined, a detailed functional analysis of the spacer acquisition machinery is lacking.

Here, we develop a reporter for the selection of cells that acquire new spacers into the *Streptococcus pyogenes* type II-A CRISPR-Cas system, both through the expression of an antibiotic resistance gene or a fluorescent protein. We use this selection in combination with a library of *cas1*, *cas2* or *csn2* variants generated by deep mutational scanning (DMS)[37] to isolate mutations that enhance spacer acquisition up to seven-fold and improve immunity against phage predation. In addition, DMS results reveal key Cas1-Cas2 interactions required for spacer acquisition by the type II-A CRISPR-Cas system. Our findings offer a platform to improve CRISPR-Cas-based technologies that rely on spacer integration that are compatible with downstream applications that utilize the type II-A RNA-guided nuclease Cas9.

## Results

### Spacer acquisition reporter for type II-A CRISPR-Cas system

Spacer acquisition of the *S. pyogenes* type II-A CRISPR-Cas system leads to the addition of 66 base pairs to the genome[20] composed of a canonical 30-nt spacer and a 36-nt repeat[13]. This added sequence length, being a multiple of three nucleotides, prevents the use of frame-shift strategies as have been described for the selection of cells with newly acquired spacers in the type I-E CRISPR-Cas system[38,39]. We therefore developed a different selection system in which the acquired spacer contains a ribosome binding site followed by an ATG start codon for the expression of a reporter gene located downstream of the CRISPR array (Fig. 1a and Supplementary Fig. 1a). To test this strategy, we transformed the laboratory strain *Staphylococcus aureus* RN4220 (ref. 40) with a chloramphenicol-resistant (Cm^R) plasmid harboring a modified *S. pyogenes* SF370 type II-A CRISPR-Cas locus[41] that contains a single repeat followed by an mNeonGreen (mNG) gene. No mNG expression was observed by live-cell fluorescence microscopy of staphylococci carrying this plasmid (Fig. 1b). However, when we engineered a spacer with a translation start site, cells displayed green fluorescence, which was eliminated when the ATG start codon was replaced by TAG (Fig. 1b, Supplementary Fig. 1b and Supplementary Fig. 2, which provides all the source images presented in this study). Fluorescence, however, was not strong and therefore we hypothesized that pre-crRNA processing by tracrRNA and RNase III (ref. 18) could potentially cleave the reporter transcript (Supplementary Fig. 1c), leading to lower levels of translation and mNG signal. To overcome this, we introduced mutations that suppress this cleavage (Supplementary Fig. 1c, d). While signal strength and homogeneity were only modestly improved with a Δ*rnc* strain, mutations in the tracrRNA designed to prevent hybridization with the repeat and thus pre-crRNA processing significantly increased the mNG signal (Fig. 1b). From these, we selected the *cs* variant for further reporter experiments due to its minimal changes to the tracrRNA and robust mNG signal. In addition, we deleted the promoter responsible for the transcription of the long form of the tracrRNA to increase expression of the *cas* operon[19] and enhance spacer acquisition[22].

We tested this spacer acquisition reporter to select cells that incorporate a new spacer after electroporation of short dsDNA prespacers that contain a translation start site (PS1 and PS2, Supplementary Fig. 1e) into a strain carrying a single CRISPR repeat and a downstream erythromycin resistance gene (*ermC*) instead of mNG. In this reporter assay, acquisition of the spacer sequences within PS1 or PS2 would lead to the formation of an erythromycin-resistant (Erm^R) colony, enabling quantification of the spacer acquisition rate by enumerating the resistant colony-forming units (CFUs) and normalizing to the total number of CFUs (Erm^R/Cm^R). Calculation of this value after electroporation of PS1 showed a high spacer acquisition rate, which

decreased to mock-electroporation levels in the absence of *csn2* or the ATG start codon in the prespacer oligonucleotides (Fig. 1c). We extracted the reporter plasmids from a subset of Erm^R colonies formed in these control experiments, amplified the CRISPR array and subjected the PCR products to Sanger sequencing. We found that these plasmids contained spacers from translation start sites of plasmid or chromosomal genes, which were integrated in-frame with the reporter gene (Supplementary Fig. 1f), a result that underscores the stringency of the selection system.

### Deep mutational scanning of Cas1, Cas2 and Csn2

We decided to employ our spacer acquisition reporter to select for amino acid changes on the type II-A supercomplex that would increase spacer acquisition rates. To do this, we performed DMS[37] to assess the functional consequences of all possible amino acid substitutions in Cas1, Cas2 and Csn2. We generated separate libraries for each gene using a PCR-based codon mutagenesis protocol[42], which ensures comprehensive sampling of all possible amino acid changes, unlike single-nucleotide mutagenesis techniques such as error-prone PCR. The libraries were subcloned into the *ermC* reporter plasmid, each of them containing, on average, 1.8, 1.3 and 1.5 mutated codons for Cas1, Cas2 and Csn2, respectively. The number of mutated codons per clone roughly followed the expected Poisson distribution with the mutations evenly distributed across the genes, confirming the quality and uniformity of the libraries (Supplementary Fig. 3). Each library was screened separately by electroporating PS1 and plating on erythromycin-containing agar plates to select for Erm^R cells that have acquired a spacer, and on chloramphenicol-containing agar plates as the library control (Fig. 2a). Comparative next-generation sequencing[37,42,43] of the two populations quantified the impact of each residue change on spacer acquisition. Normalized enrichment ratios (calculated as reads per million, RPM, for the different codons at a particular amino acid position, obtained in the sequencing of Erm^R colonies relative to the same value obtained from Cm^R colonies) indicate amino acid preferences at each position[37]. Enrichment ratios of each codon relative to the wild-type RPM values are reported as differential selection and indicate residues that are positively or negatively selected (favored or disfavored, respectively) over the wild-type residue[43]. The entire datasets are presented in Supplementary Figs. 4–10 and Supplementary Data 1.

We validated our results by analyzing the DMS results for residues known to be important during spacer acquisition. Catalytic residues in the Cas1 active site (E149, H205, D217, E220 and R223)[29,44] were strongly preferred (Supplementary Fig. 11a) and their substitutions were highly disfavored (Supplementary Fig. 11b). Interestingly, several other residues in the active site (D144, N147, G202, N210, N213 and D227) showed high preference (Supplementary Fig. 11a). We obtained an AlphaFold3 (ref. 45) model of the *S. pyogenes* Cas1-Cas2 complex and, based on this structure, we concluded that the preference of these side chains is likely due to their role in substrate binding or as secondary-shell residues essential for catalysis (Supplementary Fig. 11c). More globally, strong negative selection against most residue changes indicated stringent purifying selection pressure across the three proteins (Supplementary Figs. 4, 8–10).

We further analyzed mutational tolerance by calculating the Shannon entropy for each protein position[42,46]. As expected, invariable positions tended to cluster within the Cas1 protein core, likely due to structural reasons, whereas Cas1 surface residues exhibited greater mutational tolerance (Supplementary Fig. 11d). An exception to this trend was found in the surface-exposed residues H12, K14 and K25 of Cas1, which were highly invariable, undoubtedly due to their critical role in recruiting Cas1 to Csn2 during spacer acquisition[34–36] via electrostatic interactions with Csn2 D9, D209 and D211 (Supplementary Fig. 11d–g). Additionally, substitutions for proline residues were consistently among the most disfavored across all sites, likely due to their

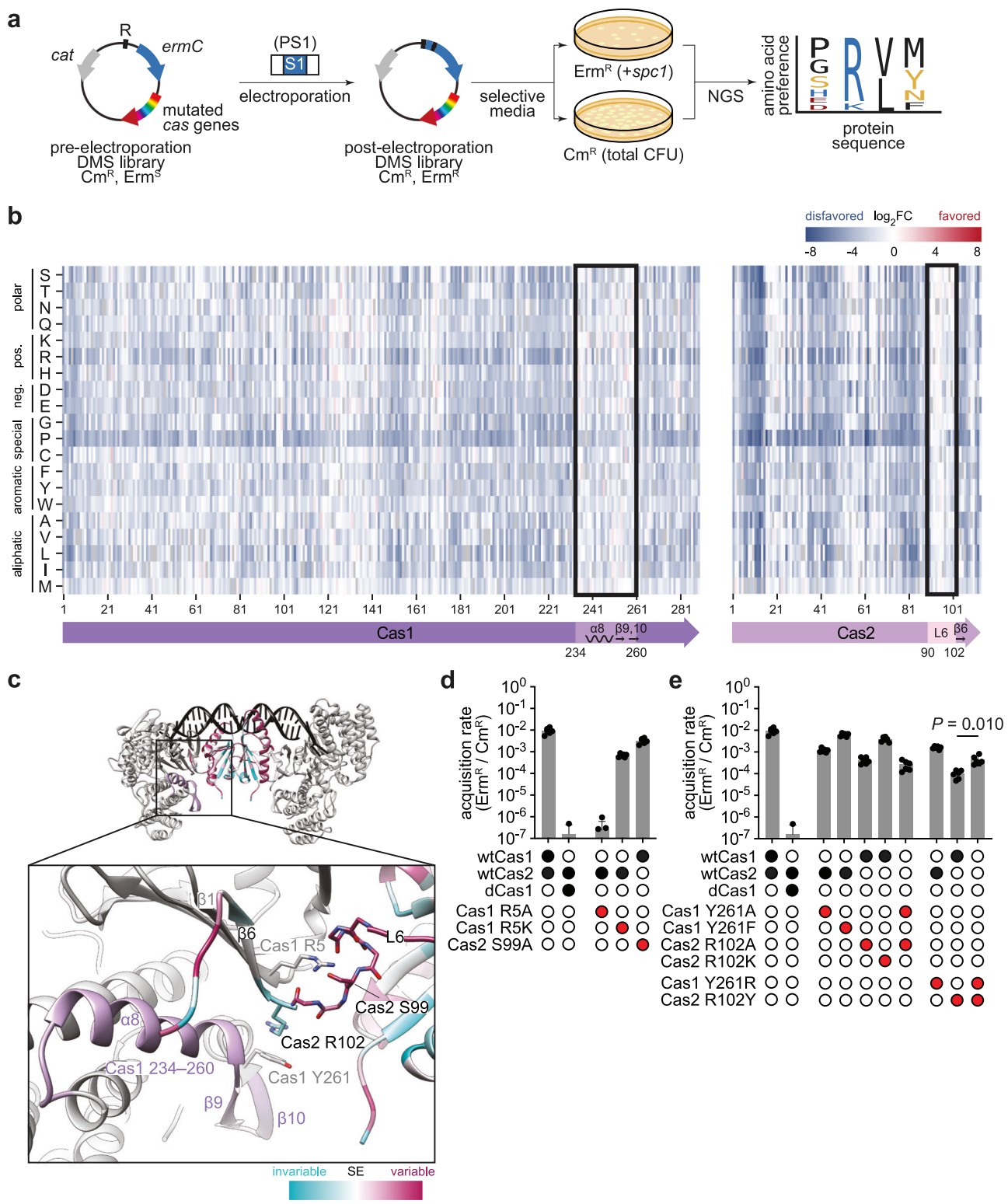

**Fig. 2 | DMS reveals interacting residues at the Cas1-Cas2 interface. a** Deep mutational scanning (DMS) workflow. Libraries were electroporated with PS1 and plated on erythromycin to select for clones that have acquired a spacer (Erm$^R$, +*spc1*), and on chloramphenicol as the unselected control (total CFU). Amino acid preferences were determined by comparative next-generation sequencing (NGS) of the two populations. **b** Differential selection of amino acids (rows) across the Cas1 and Cas2 proteins (columns), colored by log$_2$ fold change (log$_2$FC). Residues 234–260 of Cas1 and 90–102 of Cas2 are highlighted, as well as their secondary structures. **c** AlphaFold3 model of Cas1-Cas2 integrase complex. A 22-bp prespacer with 4-nt overhangs is shown in black, Cas1 dimers are shown in gray, and the Cas2 dimer is colored by Shannon Entropy (SE). The inset shows the side chains of Cas1 R5, Y261 and Cas2 S99, R102, and the backbone atoms of Cas2 residues 95–101. Cas1 residues 234–260 are highlighted in purple. **d**, **e** Spacer acquisition rate as determined by the ratio of CFUs on erythromycin and chloramphenicol (Erm$^R$/Cm$^R$) after electroporation of PS1 into reporter strains with indicated *cas* variants. Individual data points are shown with error bars representing the mean ± s.d. (technical replicates, *n* = 3 for dCas1, *n* = 6 for all other conditions).

disruptive effect on protein secondary structures (Supplementary Figs. 4, 8–10). Based on these results, we conclude that DMS analysis is a suitable approach for the identification of important residues for spacer acquisition in type II-A CRISPR-Cas systems as well as favorable and unfavorable substitutions that affect this process.

## DMS identifies critical residues at the Cas1-Cas2 interface

DMS analysis showed that residues 234–260 of Cas1 (Supplementary Fig. 12a) and residues 90–102 of Cas2 (Supplementary Fig. 12b) have relatively weak amino acid preference (Fig. 2b). Moreover, several residue changes within these regions were positively selected (Supplementary Fig. 12a, b). In an AlphaFold3 model of the integrase complex, these variable regions map to the α8-helix and β9,10-strands of Cas1 and loop 6 of Cas2, which connects the core to the β6-strand in its C-terminal tail (Fig. 2b, c). Although the *S. pyogenes* Cas1-Cas2 structure has not been determined, biochemical studies revealed an essential role of the Cas2 C-terminus (residues 92–113) for the interaction with Cas1 in the *S. pyogenes* integrase complex[34]. In addition, structural studies of related type II-A systems in *Enterococcus faecalis*[10] and *Streptococcus thermophilus*[36] showed that the C-terminal β6-strand of Cas2 interacts extensively with Cas1, forming a β-sheet with its N-terminal β1-strand. Similarly, alterations in the Cas2 C-terminus of the *E. coli* type I-E system disrupted Cas1-Cas2 complex formation and abolished spacer acquisition, underscoring the essential role of a structured Cas2 C-terminus[29]. Therefore, based on these previous observations and our structural model, the variability of residues 234–260 of Cas1 and residues 90–102 of Cas2 we measured via DMS was intriguing.

To validate our findings, we introduced amino acid substitutions in these regions and tested their impact on spacer acquisition. Our structural model predicted the formation of hydrogen bonds between the backbone amides of residues 98–101 in loop 6 of Cas2 and Cas1 R5 (Fig. 2c). R5 is located in the β1-strand of Cas1 (Fig. 2c) and displayed a strong preference score in our DMS experiments (Supplementary Fig. 12a). In agreement with this, substituting Cas1 R5A completely abolished spacer acquisition, similar to a catalytically dead Cas1 variant harboring H205A and E220A substitutions (dCas1)[26,28], whereas the conservative substitution R5K was tolerated (Fig. 2d). None of these mutations affected the formation of the Cas1-Cas2 integrase complex since Cas1 and Cas2 co-eluted as a higher-molecular-weight species in analytical size-exclusion chromatography (SEC) (Supplementary Fig. 12c). On the other hand, mutation of Cas2 S99 to alanine minimally affected spacer acquisition (Fig. 2d), corroborating that Cas2 loop 6 interactions involve mainly the peptide backbone rather than specific side chains, and explaining the observed tolerance of amino acid substitutions in this region.

The only strongly preferred residue in loop 6 was Cas2 R102 (Supplementary Fig. 12b), which forms a cation-π interaction with Cas1 Y261 in our AlphaFold3 model (Fig. 2c). Substituting Cas1 Y261A or Cas2 R102A, individually or in combination, reduced the spacer acquisition rate, while the conservative substitutions Cas1 Y261F and Cas2 R102K were tolerated (Fig. 2e). To test the interaction between these residues we swapped them. The Cas1 Y261R mutation alone displayed a minor effect on spacer acquisition (Fig. 2e). In contrast, the Cas2 R102Y substitution generated a reduction in the incorporation of new spacers by approximately two orders of magnitude, which was partially rescued in the double mutant (Fig. 2e). This result supports the existence of a specific interaction between Cas1 Y261 and Cas2 R102 required for efficient spacer acquisition by type II-A CRISPR-Cas systems.

In summary, our DMS analysis revealed that despite overall tolerance to amino acid substitution of residues 234–260 of Cas1 and residues 90–102 of Cas2, linchpin interactions between specific residues within these regions are essential for the functional integrity of the Cas1-Cas2 integrase complex. High conservation of these residues,

Cas1 R5 and Y261 and Cas2 R102, across related streptococci confirms the importance of these interactions (Supplementary Figs. 13–15).

## Selected residue changes enhance spacer acquisition

Next, we used our DMS results to investigate whether certain positively selected amino acid changes could enhance the efficiency of this process. Such residues were characterized by low Shannon entropy and high positive differential selection (Supplementary Figs. 5–7; complete scoring in Supplementary Data 1). We introduced the eight highest-scoring substitutions in Cas1, Cas2, or Csn2 (Fig. 3a–c) as individual residue changes in the reporter plasmid and performed prespacer electroporation to quantify spacer acquisition. We observed that most substitutions resulted in no significant changes or only minor improvements in spacer acquisition rates after electroporation with PS1 (Fig. 3d–f). This modest increase was consistent with the generally weak positive selection observed at most sites (Supplementary Figs. 8–10), indicating that, while these substitutions were neutral or beneficial, they did not strongly drive increased activity on their own.

We therefore decided to combine the best performing mutations, Cas1 M77H, Cas2 Y5F, Cas2 T16Q and Csn2 S85E (Fig. 3g). We found that two of these combinations, the double mutant Cas2 Y5F/Cas2 T16Q and the triple mutant Cas1 M77H/Cas2 Y5F/Cas2 T16Q markedly improved spacer acquisition, achieving a 3.9- and 5.0-fold increase in oligo spacer acquisition, respectively. To determine whether these results are specific to PS1 or generally apply to different sequences, we repeated the assay with a different prespacer oligonucleotide, PS2, which is composed of a different prespacer sequence, flanking regions, and PAM (Supplementary Fig. 1e). Although acquisition from PS2 exhibited a lower overall rate, the relative activity of the variants remained consistent (Fig. 3h), confirming that the enhancements are due to a general increase in activity rather than specialization for a particular sequence.

To test whether spacer acquisition variants identified in *S. aureus* exhibit similar enhancements in a different host, we introduced the Cas2 T16Q (designated as T), Cas2 Y5F/Cas2 T16Q (YT) and Cas1 M77H/Cas2 Y5F/Cas2 T16Q (MYT) variants into *E. coli* K-12 MG1655 (ref. 47). These variants were expressed on a tetracycline-resistant (Tet^R) reporter plasmid containing a downstream kanamycin resistance gene (*kanR*), and a modified version of PS1 that carries an *E. coli* specific ribosome binding site, PS3 (Supplementary Fig. 1e), was introduced via electroporation. In contrast to the improvement observed in *S. aureus*, variant T showed no significant enhancement in the *E. coli* host. However, variants YT and MYT showed modest improvements, increasing spacer acquisition rates by 1.7- and 1.6-fold, respectively (Fig. 3i).

## Spacer acquisition variants improve phage immunity

Finally, we tested the performance of the favorable residue changes we identified through DMS in their ability to improve spacer acquisition, and therefore the type II-A CRISPR-Cas immune response, during phage infection. Variants T, YT and MYT were introduced into a version of the reporter plasmid that harbors the *S. pyogenes* type II-A CRISPR-Cas loci with a single repeat but without the reporter gene[24]. This plasmid, along with a control plasmid lacking the spacer acquisition genes (Δ*cas1/cas2/csn2*), was transformed into staphylococci before infection with the lytic phage ΦNM4γ4 (ref. 28) in soft agar plates. In this assay, bacteria resistant to phage predation form colonies that can be enumerated and analyzed to determine the extent of spacer acquisition[21]. The total number of surviving colonies was notably higher in the presence of the three variants we tested than in the presence of wild-type Cas proteins or in their absence (Fig. 4a). Twelve colonies from each experiment were analyzed via PCR of the CRISPR array to determine the presence of new spacers. None of the colonies derived from infection of cells containing the control plasmid

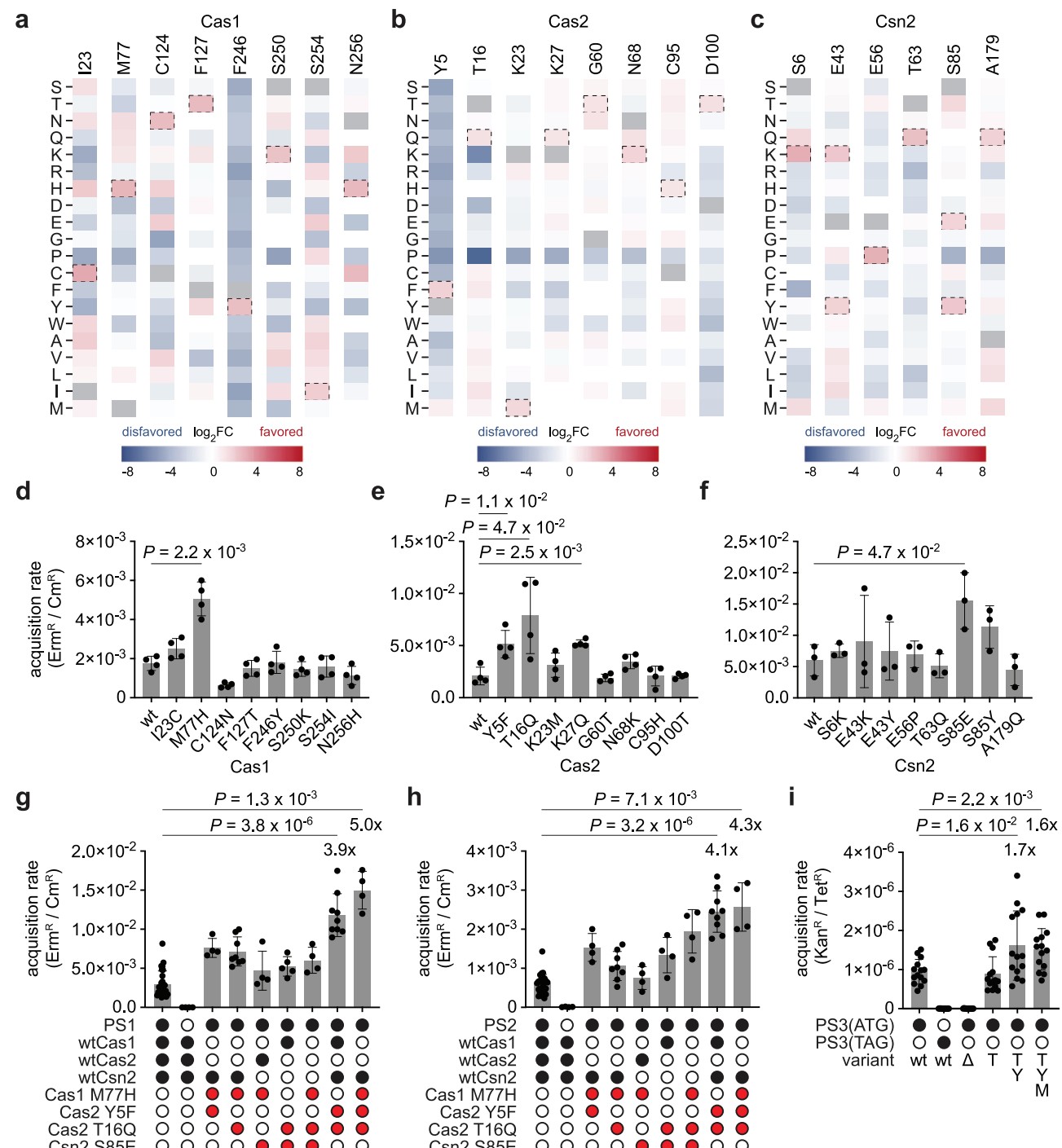

**Fig. 3 | Positively selected residue changes result in increased spacer acquisition rate.** **a**–**c** Differential selection of all possible amino acids (rows) at tested Cas1 (**a**), Cas2 (**b**) and Csn2 (**c**) sites (columns), colored by log₂ fold change (log₂FC). The amino acids introduced in the variants are marked with a dotted line. **d**–**f** Spacer acquisition rate as determined by the ratio of CFUs on erythromycin and chloramphenicol (Erm$^R$/Cm$^R$) after electroporation of PS1 into reporter strains with indicated residue changes in Cas1 (**d**), Cas2 (**e**) and Csn2 (**f**). Individual data points are shown with error bars representing the mean ± s.d. (technical replicates, $n = 4$ for Cas1 and Cas2, $n = 3$ for Csn2). Significant improvements of variants relative to wt are indicated. **g**, **h** Same as (**d**–**f**) but with

combinations of different residue changes in the reporter strain. Individual data points are shown with error bars representing the mean ± s.d. (technical replicates, $n = 4$ for all conditions, except $n = 27$ for wt, and $n = 8$ for Cas1 M77H Cas2 T16Q and Cas2 Y5F T16Q). Significance and fold changes are indicated for the two best variants. **i** Spacer acquisition rate as determined by the ratio of CFUs on kanamycin and tetracycline (Kan$^R$/Tet$^R$) after electroporation of *E. coli* reporter strains with the indicated prespacers and variants. Δ indicates deletion of *cas1*, *cas2* and *csn2*. Individual data points are shown with error bars representing the mean ± s.d. of 14 technical replicates. Significant improvements and fold changes of variants relative to wt are indicated.

displayed an expanded CRISPR array (Fig. 4b), and therefore most likely survived phage predation via non-CRISPR mechanisms, such as mutations of the viral receptor. In the presence of a wild-type type II-A CRISPR-Cas locus, 7/12 colonies added a new spacer, and this number

increased to 11/12 for the T variant and to 12/12 for the YT and MYT variants, with many colonies acquiring more than one spacer (Fig. 4b). We then quantified spacer acquisition by multiplying the total number of colonies obtained after infection by the fraction that was

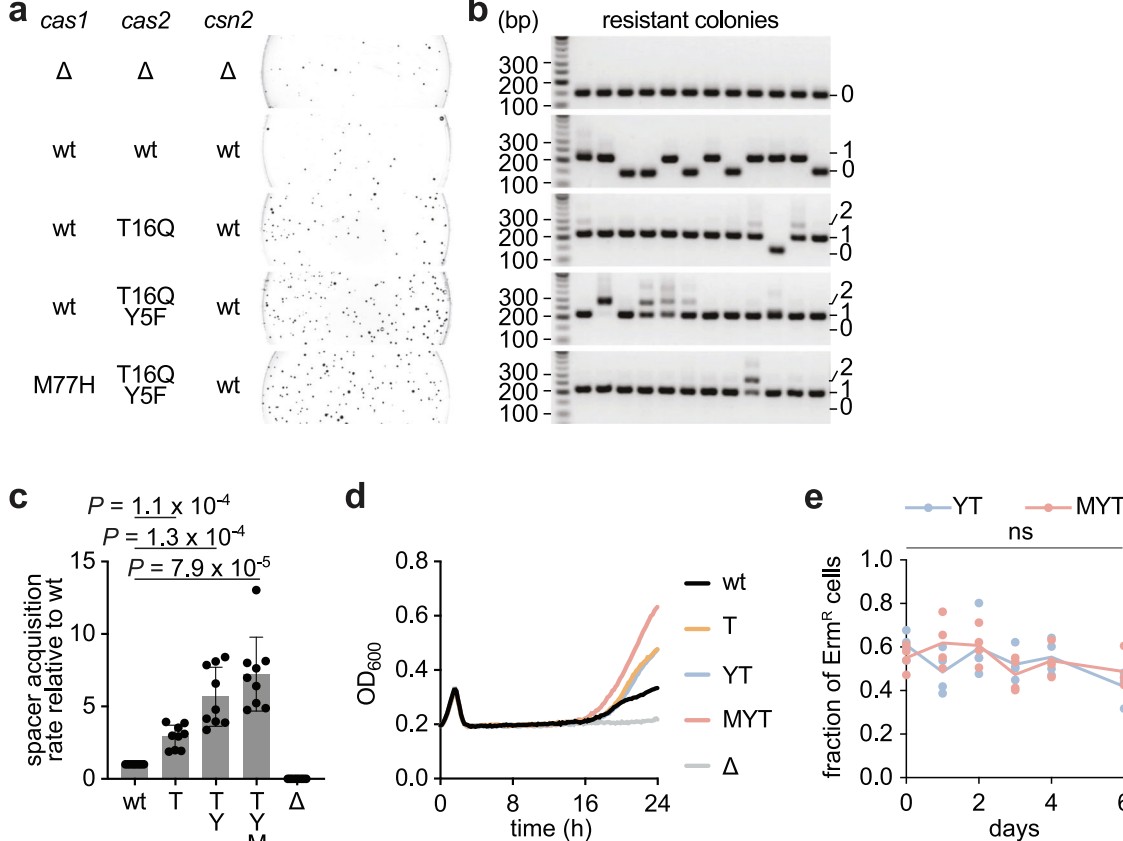

**Fig. 4 | Cas1 and Cas2 variants with enhanced spacer acquisition properties improve phage immunity without observable fitness cost. a** Plate images obtained after lytic infection of *S. aureus* strains, carrying indicated *cas* genes of the type II-A CRISPR-Cas system with a single repeat CRISPR array on a plasmid, with phage ΦNM4γ4 in soft agar. Δ indicates gene deletion. **b** Agarose gel electrophoresis of PCR amplification of the CRISPR array present in plasmids extracted from colonies that survived bacteriophage infection. The number of spacers in the PCR product (0, 1 or 2) is indicated on the right. Representative images of $n = 9$ biological replicates are shown. **c** Quantification of CRISPR-resistant colonies in **a**. Individual data points are shown with error bars representing the mean ± s.d. ($n = 9$

biological replicates). **d** Growth of strains carrying different variants of type II-A cas genes measured by $OD_{600}$ of the cultures after infection with phage ΦNM4γ4. Average data is shown ($n = 18$ individual infections from six biological replicates, each split into three wells and infected separately, with individual data shown in Supplementary Fig. 16a). **e** Fraction of $Erm^R$ staphylococci after daily passaging of mixed cultures composed of strains expressing either a wt type II-A CRISPR-Cas locus ($Cm^R$) or the indicated DMS variants ($Erm^R$ and $Cm^R$) in pairwise growth competition experiments. Individual data points are shown ($n = 4$ biological replicates). ns: not significant.

determined to harbor new spacers via PCR analysis, and divided by the wild-type value to calculate the fold improvement for each of the variants. We found approximately a 3-, 5- and 7-fold increase in acquisition for the T, YT and MYT variants, respectively (Fig. 4c). We also infected liquid cultures and followed their growth by measuring optical density at 600 nm ($OD_{600}$). In this assay, only cells that acquire a new anti-phage spacer are able to continue replicating and contribute to an increase in $OD_{600}$ detectable at approximately 16 h post-infection (Fig. 4d and Supplementary Fig. 16a–c). We found that the different variants followed a trend similar to that obtained for the spacer acquisition experiment in solid media. After 24 h, nine cultures from each condition were analyzed via PCR of the CRISPR array to test for the presence of newly acquired spacers. Expanded arrays were predominantly observed in cultures with high $OD_{600}$ values, indicating that recovery from lytic phage infection was driven by spacer acquisition (Supplementary Fig. 16b). Altogether, these findings show that the Cas variants that enhanced spacer acquisition from dsDNA oligonucleotide protospacers also mediate a better CRISPR immune response in vivo, during lytic infection.

High rates of spacer acquisition can potentially lead to CRISPR-Cas autoimmunity due to the genotoxicity caused by spurious acquisition of spacers from the host's genomic DNA or plasmids[21,22,30,48]. Indeed, the increase in spacer acquisition achieved through over-

expression of the type II-A *cas* operon leads to a fitness cost to cultures growing in the absence of phage infection[21,49]. To test this possibility, we performed pairwise growth competitions between an *S. aureus* RN4220 strain carrying the wild-type type II-A CRISPR-Cas locus and an *S. aureus* OS2 strain ($Erm^R$) (ref. [50]) harboring the same plasmid with the YT or MYT variants. Cultures were mixed at 1:1 ratios, grown in rich media overnight and plated on chloramphenicol- and erythromycin-containing agar to enumerate total and variant number of CFUs, respectively, before diluting into fresh media. The variant/total CFU ratio was plotted each day for six consecutive days (Fig. 4e). The ratio did not deviate significantly from 0.5 for any of the variants, a result that indicates that, at least under our experimental conditions, the residue changes did not incur a noticeable fitness cost. This finding is consistent with the lack of conservation for Cas1 M77, Cas2 Y5 and Cas2 T16 (Supplementary Figs. 13, 14 and 16d, e), which suggests that different residues, even if they increase spacer acquisition, are not selected against in these positions.

## Discussion
We developed an efficient selection system for type II-A CRISPR-Cas spacer acquisition that is independent from immunity against phage infection, allowing for high-throughput screening of a DMS library of *cas* genes. This approach provided insights into the sequence-function

relationships of Cas1, Cas2 and Csn2 proteins, as well as identified Cas1 and Cas2 variants that support up to 5-fold increase in spacer acquisition and 7-fold increase in anti-viral defense.

The DMS data showed predominantly strong negative differential selection across Cas1, Cas2 and Csn2 protein sequences, indicating a constrained evolutionary space for permissible residue changes (Supplementary Figs. 8–10). This constraint likely stems from the complex nature of type II-A spacer acquisition, which involves multiple stages and conformational changes: formation of a prespacer capture supercomplex, processing and transfer of the prespacer to the Cas1-Cas2 integrase, and subsequent cleavage-ligation reactions at the leader-proximal end of the CRISPR array to integrate the new spacer[10,34–36]. Each of these steps may depend on distinct protein interactions and enzymatic functions. Another constraint stems from the formation of homodimers and higher-order multimers by Cas1, Cas2 and Csn2 (refs. 32,35,51), with each protomer occasionally performing distinct functions within a single enzymatic step. An example of this complexity is the variable α8-helix of one Cas1 protomer (Cas1a), which we analyzed for its interaction with the β6-strand of Cas2 (Fig. 2c), that assumes a different structural context in the other Cas1 protomer (Cas1b) of the integrase complex predicted by Alpha-Fold3. Unlike in Cas1a, the α8-helix of Cas1b is located away from the Cas2 binding interface and instead contacts the prespacer DNA (Supplementary Fig. 17a). Likewise, interaction between the Cas1 dimer and Csn2 in the spacer capture complex is not symmetric, with each Cas1 protomer interacting with different residues of Csn2 (Supplementary Fig. 11d–g)[35,36].

In contrast to the strong overall negative differential selection, we examined two regions at the Cas1-Cas2 interface that displayed relatively low amino acid preferences. One region corresponds to Cas2 loop 6, which, in the AlphaFold3 model, primarily interacts through the peptide backbone, forming a key interaction with the side chain of Cas1 R5. Substitution of R5 to alanine completely abolished spacer acquisition. Given that residue changes at this position did not disrupt Cas1-Cas2 integrase assembly (Supplementary Fig. 12c), R5A may instead interfere with the assembly of the type II-A supercomplex or with some of its functions in spacer sampling and integration. The second variable region is the α8-helix of Cas1, which is positioned for potential interactions with the Cas2 β6-strand (Fig. 2c, Supplementary Fig. 17b). The α8-helix exhibits strong amphipathic character, with solvent-exposed positions enriched in poorly conserved polar residues, while relatively conserved hydrophobic core residues pack with neighboring Cas1 α6- and α7-helices (Supplementary Fig. 17c, d). Interactions with Cas2 are primarily mediated by non-specific van der Waals interactions, with the α8-helix acting as a hydrophobic platform for the Cas2 L103 and F105 side chains of the β6-strand (Supplementary Fig. 17b). The only polar residue in the platform, K242, is highly conserved but does not interact with Cas2; instead, it is stabilized by E44 in the Cas1 β5-strand. Overall, the partially exposed surface of the α8-helix, combined with its reliance on non-specific hydrophobic interactions with Cas2, contribute to its substantial tolerance for various residues.

Given the complexity of the spacer acquisition process and the challenges associated with rationally predicting residue changes that enhance functionality, our DMS approach, coupled with a functional selection, proved effective in identifying some residue changes that improved activity and, consequently, enhanced anti-phage immunity. While only a subset of residue changes significantly increased spacer acquisition rates on their own—likely a result of the weak positive differential selection—certain combinations of these changes resulted in substantially improved variants. Based on these findings, shuffling of positively selected alleles could be a fruitful strategy to more comprehensively identify additive or synergistic effects that further increase spacer acquisition efficiency, as well as further rounds of diversification and selection.

One of the most effective residue changes we identified was the relatively conservative Cas2 Y5F. Structural models of the Cas1-Cas2 integrase complex suggest that both Cas2 Y5 and Y5F contribute to the structural organization of the variable Cas2 loop through several van der Waals interactions—along with the analyzed cation-π interaction between Cas1 Y261 and Cas2 R102 and the backbone interactions of Cas1 R5 (Supplementary Fig. 18a, b). Analytical SEC showed that both Cas2 Y5F and Y5A formed higher-molecular-weight complexes with Cas1 (Supplementary Fig. 18c), similar to the wild-type (Supplementary Fig. 12c), indicating that these residue changes do not affect integrase complex assembly. Instead, the enhanced activity of Cas2 Y5F may result from allosteric modulation of catalysis or structural effects at a different stage of spacer acquisition. We also identified the Cas1 M77H substitution, located at the interface between the N-terminal β-sandwich-like domain (residues 1–81) and the C-terminal α-helical domain (residues 88–289)[35]. Interestingly, directed evolution of E. coli Cas1 for enhanced spacer acquisition also produced a variant with the residue change V76L at the same interface[39]. This E. coli Cas1 residue corresponds to the homologous I66 residue in S. pyogenes Cas1, positioned in an adjacent β-strand to M77 (β7-strand for I66, β8-strand for M77) (Supplementary Fig. 18d). Therefore, both the E. coli Cas1 V76L and S. pyogenes Cas1 M77H changes are expected to impact the relative motion of the Cas1 domains, which is mediated by a flexible linker (residues 82–87) during spacer integration[10,35,39]. Finally, Cas2 T16Q is located in proximity to the dsDNA prespacer in the AlphaFold3 model (Supplementary Fig. 18e, f). We believe that this residue change could influence DNA binding over the course of spacer acquisition.

The introduction of spacer acquisition variants into E. coli revealed notable differences in performance compared to S. aureus. Overall, the ability of the variants to enhance spacer acquisition was weaker in E. coli, with variant T exhibiting activity comparable to the wild-type system, while the YT and MYT variants resulted in only modest levels of improvement. These findings suggest that host-specific factors affect spacer acquisition, which is consistent with prior observations[25,52,53]. In spite of this, our results demonstrate partial transferability of the insights obtained in the S. aureus screen into E. coli, and establish the reporter system in E. coli to conduct a similar screen directly in this host. To our knowledge, type II-A spacer acquisition has not been previously demonstrated in E. coli.

The residue changes identified through DMS that enhanced spacer acquisition and phage immunity were not highly conserved in related natural sequences (Supplementary Figs. 13–15 and 16d, e), likely due to the presence of selection pressures at play other than host defense. One possibility is that, despite the high rate of phage predation, bacteria also rely on a range of other immune mechanisms besides CRISPR-Cas[54], which, when present in the same host, could weaken the selective pressure for more efficient CRISPR-Cas spacer acquisition[55]. Another possibility, which we tested in this study, was a fitness cost associated with the promotion of self-spacer acquisition in cells expressing the identified Cas1-Cas2 variants. However, we did not observe such a fitness cost. This is in contrast to our previous analysis of the hyper-acquisition Cas9 I473F variant, which was outcompeted by wild-type Cas9 (ref. 21). We believe that this difference is a result of the contrasting effects of the variants. On one hand, the Cas9 I473F variant affects the binding and function of the long form of the tracrRNA, preventing auto-repression of the cas operon and triggering the overexpression of the type II-A cas operon, which increases spacer acquisition and Cas9 targeting by approximately 100-fold (ref. 22). Such increase could indeed impact self-acquisition rates and lead to genotoxicity. On the other hand, our work built on the cas-overexpression system and allowed an improvement of 5–7 fold over the increase generated by cas-overexpression. This more moderate increase in spacer acquisition could still mediate a mild genotoxic effect, potentially leading to subtle fitness costs that might only become detectable over a longer time frame or under different growth

and environmental conditions. Another possible explanation for the absence of a fitness cost in our pairwise competition experiment, conducted without phage infection, is that rapid and repeated spacer acquisition from the same phage might disadvantage the host by compromising long-term immunity against a broader range of phages. The identified high-acquisition variants offer an opportunity to explore these hypotheses further.

We believe that the Cas1-Cas2 variants we identified will be useful for technologies that rely on spacer acquisition such as the CRISPR-based protection of industrial bacterial cultures from phage predation[56] and the recording of biological signals into stable genetic information[57] in applications that use the insertion of spacers into the CRISPR array to register the temporal availability of metabolites[58], electrical stimuli[59], horizontal transfer events in the gut microbiota[60] and gene expression[61,62]. A major constrain of technologies based on spacer acquisition with wild-type systems is the low frequency of the process, which occurs in as few as 1 in $10^7$ cells[28]. The increase we obtained is similar to that reported by a recent study that evolved the *E. coli* type I-E Cas1-Cas2 complex (4–10 fold) (ref. 39). We believe that an advantage of our approach is the possibility of providing new spacers for the generation of RNA guides for Cas9. The Cas1-Cas2 variants we identified could be linked to downstream synthetic biology applications that employ this nuclease, for example the construction of gene repression libraries[63].

## Methods

### Bacterial strains and growth conditions

*S. aureus* RN4220 (ref. 40) and OS2 (ref. 50) were cultured in brain heart infusion (BHI) broth at 37 °C and 220 rpm, supplemented with chloramphenicol (10 µg mL⁻¹) or spectinomycin (250 µg mL⁻¹) to maintain pC194-(ref. 64) and pLZ12- (ref. 65) based plasmids, respectively. To select for *S. aureus* OS2, media was additionally supplemented with erythromycin (10 µg mL⁻¹). *E. coli* EC100, K-12 MG1655 (ref. 47) and Rosetta 2(DE3) BL21 (Novagen) were cultured in lysogeny broth (LB), unless otherwise specified, supplemented with kanamycin (50 µg mL⁻¹) to maintain pET-based plasmids, and with tetracycline (10 µg mL⁻¹) to maintain pACYC184-based plasmids. Cultures of *E. coli* Rosetta 2(DE3) BL21 were additionally supplemented with chloramphenicol (12.5 µg mL⁻¹) to maintain pRARE2.

### Bacteriophage propagation

To make a high-titer stock of phage ΦNM4γ4 (ref. 28), an overnight culture of *S. aureus* RN4220 was diluted 1:100 in fresh media supplemented with 5 mM CaCl₂. The culture was grown to mid-log phase and infected with phage at MOI 0.1. The infection was allowed to proceed for 4 h at 37 °C and 220 rpm. The lysate supernatant was obtained by centrifugation (4000 × *g*, 10 min, 4 °C) and passed through a sterile membrane filter (0.45 µm). The phage stock was stored at 4 °C. Phage titers were determined by serially diluting the phage stock tenfold in BHI, followed by spotting of 3–5 µL of the dilutions onto 50% heart infusion agar (HIA), mixed with 5 mM CaCl₂ and 100 µL of an *S. aureus* RN4220 overnight culture, layered onto a plate of BHI agar. The plates were incubated overnight at 37 °C, and PFUs were counted the next day.

### Molecular cloning

Plasmids, with details on their construction and the oligonucleotide primers used, are listed in Supplementary Data 2 and 3, respectively.

### Strain construction

To create *S. aureus* RN4220 Δ*rnc* (*rnc::aadA*), Gibson assembly was used to join the PCR amplicons of an upstream homology fragment (oCH9 and oCH10), a downstream homology fragment (oCH13 and oCH14), both from genomic DNA, and an *aadA* spectinomycin resistance cassette (oCH11 and oCH12). The linear Gibson product was transformed into electrocompetent *S. aureus* RN4220 harboring the recombineering plasmid pPM260 (ref. 66) and selected for with spectinomycin (150 µg mL⁻¹). Potential integrants were screened by colony PCR and verified by Sanger sequencing.

### Fluorescence microscopy

Overnight cultures of *S. aureus* RN4220 or RN4220 Δ*rnc* harboring pCH2, pCH4, or pCH6–pCH21 were spun down, resuspended in PBS (Gibco) and spotted onto 1% agarose in PBS. Live-cell images were acquired using a Nikon Ti-E motorized inverted microscope equipped with Perfect Focus System, a CFI60 Plan Apochromat DM Lambda 100X oil objective lens (Nikon, NA 1.4), a SOLA Light Engine (Lumencor), and a Zyla 4.2 sCMOS camera (Andor, pixel size 65 nm). Phase contrast was imaged with the SOLA Light Engine set to 2% power with 10 ms excitation. mNG was imaged with the SOLA Light Engine set to 2% power with an exposure time of 1500 ms, using A C-FL GFP HC HISN Zero Shift filter (excitation: 470/40 nm (450–490 nm), emission: 525/50 nm (500–550 nm), dichroic mirror: 495 nm). The microscope was controlled by the software NIS-Elements AR version 5.21.03. Image analysis was done with ImageJ2's Fiji v2.3.0.

### Spacer acquisition from dsDNA oligonucleotides

Highly electrocompetent *S. aureus* RN4220 cells carrying pRAH49-1, pRAH50, pRAH77, or variants thereof with indicated residue changes, were obtained by diluting an overnight culture 1:100 in fresh media supplemented with 0.5 M sorbitol (BHIS). Cultures were grown to an OD₆₀₀ of 0.6–0.8 before the cells were washed with 0.5 M sucrose. Washed cells were resuspended in 0.5 M sucrose, and the cell density was normalized to an OD₆₀₀ of 25. Highly electrocompetent *E. coli* K-12 MG1655 cells carrying pRAH131 or pRAH276–pRAH279 were obtained by diluting an overnight culture 1:100 in fresh media. Cultures were grown to an OD₆₀₀ of 0.5–0.7 before the cells were washed with 10% v/v aq. glycerol. Washed cells were resuspended in 10% v/v aq. glycerol, and the cell density was normalized to an OD₆₀₀ of 50. Aliquots of competent cells were stored at −80 °C until use. Electroporation of dsDNA oligonucleotides was carried out similarly as described by Heler et al.[28] Briefly, ssDNA oligonucleotide primers were dissolved at 1 mM concentration and annealed in Duplex Buffer (IDT). The annealed primer pairs were oRAH75 and oRAH76 for PS1(ATG), oRAH167 and oRAH168 for PS1(TAG), oRAH384 and oRAH385 for PS2, oRAH501 for PS3(ATG), and oRAH502 and oRAH503 for PS3(TAG). Following dialysis against water, 5 or 10 µg dsDNA oligonucleotides were transformed into 50 µL highly electrocompetent *E. coli* K-12 MG1655 or 100 µL highly electrocompetent *S. aureus* RN4220, respectively, harboring indicated reporter plasmids. Cells were recovered in 900 µL BHIS (*S. aureus*) or 450 µL SOC media (Corning) (*E. coli*) at 37 °C and 220 rpm for 2.5 h. *S. aureus* transformations were serially diluted tenfold in BHI and plated on BHI agar containing chloramphenicol (10 µg mL⁻¹) to quantify total CFU, and on BHI agar containing erythromycin (10 µg mL⁻¹) to quantify the number of cells that have acquired a spacer. *E. coli* transformations were serially diluted tenfold in LB and plated on LB agar containing tetracycline (10 µg mL⁻¹) to quantify total CFU, and on LB agar containing kanamycin (10 µg mL⁻¹) to quantify the number of cells that have acquired a spacer. CFUs were counted after incubating the plates at 37 °C for approximately 16 h (*E. coli*) or 20 h (*S. aureus*). Technical replicates were conducted by electroporating individual aliquots of the corresponding reporter strains.

### Spacer acquisition in soft agar

Overnight cultures of *S. aureus* RN4220 harboring pRAH225–pRAH229 were diluted 1:100 in fresh media. Cultures were incubated at 37 °C and 220 rpm for 3 h. OD₆₀₀ was measured to determine the cell density, 2.8 × 10⁸ cells were mixed with phage at an MOI of 2 in 5 mL of 50% HIA top agar with 5 mM CaCl₂, and the mixture was layered onto BHI agar plates to solidify at rt (30 min). Top agar and plates were

supplemented with appropriate antibiotics to maintain the plasmids. The plates were incubated overnight at 37 °C and BIMs were counted the next day. To quantify spacer acquisition, individual BIMs were restreaked and DNA was isolated by resuspending colonies in 30 µL of lysis buffer (50 mM Tris-HCl pH 9.0, 250 mM KCl, 5 mM MgCl$_2$, 0.5% v/v Triton X-100) supplemented with lysostaphin (AMBI Products, 0.1 mg mL$^{-1}$ final concentration). Lysates were incubated at 37 °C for 20 min and at 98 °C for 10 min. Samples were centrifuged and the supernatant was diluted tenfold in water. Of the dilution, 1 µL was used as the template for PCR amplification with Phusion polymerase (ThermoFisher) using oRAH56 and oRAH59 in 25 µL reaction volume. The thermocycler settings were: 98 °C for 30 s, 30 cycles of [98 °C for 10 s, 65 °C for 10 s, 72 °C for 20 s], 72 °C for 20 s, and hold at 12 °C. PCR products were analyzed on 2% agarose gels and visualized with ethidium bromide. Plates and agarose gels were imaged using an Amersham ImageQuant 800 (Cytiva). Images were adjusted for brightness and contrast with ImageJ2's Fiji v2.3.0. Biological replicates of soft agar infections were conducted by picking individual colonies for overnight cultures. For each replicate and condition, 16 BIMs were analyzed by PCR, except for the Δ*crispr* condition, where 8 BIMs were analyzed.

### Spacer acquisition in liquid culture

Overnight cultures of *S. aureus* RN4220 harboring pRAH225–pRAH229 were diluted 1:100 in fresh media supplemented with 5 mM CaCl$_2$. Cultures were incubated at 37 °C 220 rpm for 1.5 h and normalized to OD$_{600}$ of 0.3. Phage ΦNM4γ4 was added to 150 µL of the normalized cultures in 96-well plates, and OD$_{600}$ was measured every 10 min in a microplate reader (TECAN Infinite 200 PRO) at 37 °C with shaking. To test for spacer acquisition, cultures were spun down after 24 h. The cell pellets were resuspended in 150 µL of resuspension buffer (Qiagen) supplemented with lysostaphin (0.1 mg mL$^{-1}$ final concentration). Following incubation at 37 °C and 220 rpm for 30 min, plasmids were isolated using MiniPrep kits (Qiagen) and eluted in 30 µL H$_2$O. PCR reactions with oRAH56 and oRAH59 were performed and analyzed as described above for spacer acquisition in soft agar, using 1 µL of isolated plasmids as template. Biological replicates were conducted by picking individual colonies for overnight cultures, and each biological replicate was split into three wells and infected separately.

### Growth competition

Plasmids pRAH226, pRAH228 and pRAH229 were transformed into *S. aureus* RN4220 (no antibiotics resistance) and OS2 (erythromycin resistance). Overnight cultures were launched from single colonies and diluted the next day 1:100 in fresh BHI with spectinomycin (250 µg mL$^{-1}$). Cultures were grown for 1 h, their cell densities were normalized according to OD$_{600}$, and indicated combinations of strains were mixed together. The mixed cultures were passaged for 6 days by diluting them 1:100 in fresh BHI with spectinomycin (250 µg mL$^{-1}$) and then growing them overnight at 37 °C and 220 rpm. Overnight cultures were serially diluted tenfold in BHI and plated onto BHI agar containing spectinomycin (250 µg mL$^{-1}$) to quantify total CFU, and on BHI agar containing spectinomycin (250 µg mL$^{-1}$) and erythromycin (10 µg mL$^{-1}$) to quantify the number of OS2 cells. Biological replicates were conducted by picking individual colonies for initial overnight cultures.

### Codon mutant library

Codon mutant libraries were prepared following a previously described method[37]. Tiled mutagenic primers were designed using a published script (https://github.com/jbloomlab/CodonTilingPrimers). One round of mutagenesis was performed, using the end primers oRAH394 and oRAH407 for *cas1*, oRAH412 and oRAH409 for *cas2*, and oRAH392 and oRAH393 for *csn2*. The following modifications were made to the thermocycler settings: 60 s extension time was used for *cas1* and *csn2* reactions, 30 s for *cas2* reactions, and 20 cycles were

used for template amplification and the joining PCR. The library amplicons were subcloned into the ErmR reporter plasmid (amplicons of pRAH49-1 with oRAH408 + oRAH393, oRAH410 + oRAH411 and oRAH391 + oRAH394 for *cas1*, *cas2* and *csn2*, respectively) using NEBuilder HiFi DNA Assembly (NEB, E2621). The assembly reactions were dialyzed against water and transformed into electrocompetent *S. aureus* RN4220. The transformed cells were plated on BHI agar supplemented with chloramphenicol (10 µg mL$^{-1}$) and incubated overnight at 37 °C. Serial dilutions of the transformation mixtures were plated separately, which showed that we obtained 1.4 × 10$^5$ (25x library coverage, assuming one mutated codon per clone), 2.8 × 10$^5$ (125x library coverage) and 5.0 × 10$^5$ (113x library coverage) individual clones for *cas1*, *cas2* and *csn2* libraries, respectively. Controls lacking an insert yielded approximately 100-fold fewer clones. The clones were scraped from the plates, resuspended in BHI and stored as glycerol stocks at −80 °C.

### DMS library screen

Libraries were made highly electrocompetent and transformed with dsDNA oligonucleotides (PS1(ATG)) as described above for the ErmR reporter strain. A large inoculum of the library glycerol stocks was used to start the overnight cultures when making competent cells. Transformed cells were plated on BHI agar supplemented with erythromycin (10 µg mL$^{-1}$) to obtain selected libraries, and on BHI agar supplemented with chloramphenicol (10 µg mL$^{-1}$) to obtain unselected libraries. An unmutated control (pRAH49-1, wt control) was screened in the same way. Serial dilutions were plated separately, which showed that we obtained at least 3.8 × 10$^5$ individual clones for each selected and unselected library (65x, 165x and >68x library coverage for *cas1*, *cas2* and *csn2*, respectively), and 3.0 × 10$^5$ for the wt controls. Control transformations with PS1(TAG) (protospacer with a stop codon) yielded >1000x fewer clones. The clones were scraped from the plates and resuspended in BHI, and plasmids were isolated using MiniPrep kits (Qiagen). Screens were independently repeated two times (*cas1*, *cas2* and wt) or three times (*csn2*).

### Illumina sequencing

Sample preparation and comparative sequencing were performed similarly as previously described[42,46]. PCR amplicons of the full-length genes of interest were generated for each of the samples and replicates (selected and unselected *cas1*, *cas2* and *csn2* libraries, and unselected wt *cas1*, *cas2 and csn2* controls) using oRAH407 + oRAH394 for *cas1* genes, oRAH409 + oRAH412 for *cas2*, and oRAH387 + oRAH389 for *csn2*. The following modifications were made to the thermocycler settings: 60 s extension time and 20 cycles were used for all samples, and 15 ng of plasmid DNA was used as template to maximize diversity. Barcoded sub-amplicons of *cas1* genes were obtained in five segments using oRAH413 + oRAH421, oRAH422 + oRAH423, oRAH424 + oRAH425, oRAH426 + oRAH427 and oRAH428 + oRAH420, of *cas2* in two segments using oRAH429 + oRAH430 and oRAH431 + oRAH432, and of *csn2* in three segments using oRAH395 + oRAH396, oRAH397 + oRAH398 and oRAH399 + oRAH400. As template, the full-length amplicons of the two replicates of *cas1* and *cas2* screens were pooled to maximize diversity when generating the barcoded sub-amplicons. Replicates of *csn2* screens were treated independently. Pooled sub-amplicons for each sample were amplified using the forward primer Rnd2forUniv and an indexed reverse primer (cym-TruSeq1, −2, −3, −5, −8, −9 or −10). Samples were spiked with 30% PhiX and sequenced using Illumina NextSeq (mid output, paired-end 2×150-bp and 2×156-bp for *cas1/cas2* and *csn2*, respectively) with 0.45 pM loading. Data were analyzed using dms_tools2 (https://github.com/jbloomlab/dms_tools2). The Pearson correlation coefficients for amino acid preferences between pairs of *csn2* replicates were 0.81, 0.79, and 0.81, while the correlations for differential selection were 0.76, 0.75, and

0.77. For the *csn2* data, average amino acid preferences and median differential selection values are reported.

## DMS scoring

The Shannon entropy *SE*, total positive differential selection *pos_sum_diffsel*, and largest individual differential selection *max_diffsel* were linearly normalized by assigning a value from 0 for the smallest to 1 for the largest value of each category per protein. To identify potential positively selected residue changes, protein sites were scored according to: *score = pos_sum_diffsel_scaled + max_diffsel_scaled − SE_scaled* (provided in Supplementary Data 1).

## Protein expression and purification

Cas1 variants and Cas2 were expressed and purified similarly as previously described[10,67]. For Cas1 variants, overnight cultures of *E. coli* Rosetta 2(DE3) BL21 cells harboring pRAH246, pRAH248 or pRAH249 were diluted 1:100 in terrific broth (TB). Cultures were incubated in baffled shake flasks at 37 °C and 180 rpm until an $OD_{600}$ of approximately 0.6 was reached. Protein expression was induced by the addition of IPTG to a final concentration of 0.5 mM. Expressions were carried out for approximately 16 h at 18 °C and 180 rpm. Cells were collected by centrifugation (3200 x g, 25 min, 4 °C), resuspended in 20 mL lysis buffer (50 mM HEPES pH 7.6, 500 mM NaCl, 15 mM imidazole, 5% v/v glycerol) per Liter of culture and stored at −80 °C until purification. Cell suspensions were thawed at rt and placed on ice, supplemented with cOmplete mini EDTA-free protease inhibitor (Roche), 1 mM final DTT, and 0.1 mg mL⁻¹ of lysozyme. Samples were nutated at 4 °C for 1 h. Cells were lysed by sonication and the suspensions were cleared by centrifugation (16,000 x g, 45 min, 4 °C) and filtration (0.2 μm membrane filter). Supernatants were subjected to gravity-flow Ni-NTA affinity purification using binding buffer (20 mM HEPES pH 7.6, 500 mM NaCl, 15 mM imidazole, 5% v/v glycerol) and elution buffer (20 mM HEPES pH 7.5, 500 mM NaCl, 300 mM imidazole, 5% v/v glycerol, 1 mM TCEP). Samples were dialyzed against IEX buffer (20 mM HEPES pH 7.5, 5% v/v glycerol, 1 mM TCEP) with 150 mM KCl in the presence of SUMO protease (Sigma) for 18 h at 4 °C. Digested samples were purified on a HiTrap Heparin HP column using a gradient of 150 mM to 1 M KCl in IEX buffer. Fractions containing cleaved Cas1 variants were pooled, concentrated by spin diafiltration (10 kDa MWCO, Amicon), and further purified on a Superdex 200 Increase 10/300 column equilibrated with SEC buffer (20 mM HEPES pH 7.5, 500 mM NaCl, 5% v/v glycerol, 5 mM DTT). Purified samples were concentrated by spin diafiltration (10 kDa MWCO, Amicon), flash-frozen in liquid nitrogen and stored at −80 °C.

Cas2 was expressed and purified from *E. coli* Rosetta 2(DE3) BL21 cells harboring pRAH247, pRAH272 or pRAH273 using the same protocol as described for Cas1, except that dialysis against IEX buffer during SUMO protease treatment was done in the presence of 500 mM KCl, and HiTrap Heparin HP column purification was done using a gradient of 500 mM to 1 M KCl in IEX buffer.

## Analytical SEC

Analytical SEC was performed on a Superdex 200 Increase 10/300 GL column equilibrated with SEC buffer at 4 °C. Analyses were performed on an AKTA pure 25 (GE Healthcare) controlled by Unicorn v6.3. Mixtures of Cas1 variants (15–100 μM, 2 equiv.) and Cas2 variants (1 equiv.), or individual protein standards, were prepared in 200 μL final volume of SEC buffer at the indicated concentrations. Samples were incubated at 4 °C for 1 h before injection and analyzed at a flow rate of 0.7 mL min⁻¹. Fractions were analyzed by SDS-PAGE and visualized with Coomassie blue.

## Structure analysis

An AlphaFold3 (ref. 45) model of the *S. pyogenes* integrase complex was obtained with four Cas1, two Cas2, two magnesium ions, and two DNA strands forming a splayed dsDNA prespacer (5′-ATTTAGGAG-GATGATTATTTATGAAC-3′, 5′-ATAAATAATCATCCTCCTAAATTCAT-3′). An AlphaFold3 model of the *S. pyogenes* spacer capture complex was obtained with four Csn2, two Cas1 and one Cas2. Structure analyses and RSA calculations were done with PyMOL v2.4.1 and UCSF Chimera v1.18.

## Sequence alignment

The amino acid sequences of *S. pyogenes* Cas1, Cas2 and Csn2 were used as queries in a BLASTP v2.13.0 search against the NCBI RefSeq Select protein database (E-value threshold 0.05)[68]. Hits were filtered to include only sequences from non-redundant organisms that are in complete type II-A CRISPR-Cas systems. The curated sequences were aligned using CLUSTAL Omega v.1.2.4 via the EMBL-EBI web interface[69]. Sequence conservation was analyzed using Jalview v2.11.3.3[70]. Alignments of the *S. pyogenes* sequences with the ten most closely related homologs are shown in Supplementary Figs. 8–10.

## Statistics and reproducibility

Statistical analyses were performed using GraphPad Prism v10.2.3. Error bars and number of replicates are described in the figure legends. Two-tailed unpaired *t* tests with Welch's correction were used to calculate *P* values. No statistical method was used to predetermine sample size. No data were excluded from the analyses. The experiments were not randomized. The investigators were not blinded to allocation during experiments and outcome assessment.

## Reporting summary

Further information on research design is available in the Nature Portfolio Reporting Summary linked to this article.

# Data availability

Next-generation sequencing data generated in this study have been deposited in the NCBI database with BioProject ID PRJNA1212209. Source data are provided with this paper.

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

## Acknowledgements

The authors thank all the members of the Marraffini lab and Shoshanna Kahne for helpful discussions and feedback; The Rockefeller University Genomics Resource Center for next-generation sequencing; Philip Nussenzweig for plasmid pPN75; and Ailong Ke for Addgene plasmids #112793 and #112794. R.H. is supported by the EMBO Postdoctoral (ALTF 622-2020) and SNSF Postdoc.Mobility (P500PB_214377) fellowships. L.A.M. is an investigator of the Howard Hughes Medical Institute.

## Author contributions

R.H. and L.A.M. conceived the study. C.H. performed reporter optimization with tracrRNA variants. C.Y.M. and J.M. helped setting up DMS. R.H. performed all other experiments. R.H. and L.A.M. wrote the manuscript. All authors read and approved the manuscript.

## Competing interests

L.A.M. is a cofounder and Scientific Advisory Board member of Intellia Therapeutics, a cofounder of Eligo Biosciences and a Scientific Advisory Board member of Ancilia Biosciences. The remaining authors declare no competing interests.
