## [Transparent Peer Review file · Nature Communications]

Deep mutational scanning identifies Cas1 and Cas2 variants that enhance type II-A CRISPR-Cas spacer acquisition

Corresponding Author: Dr Luciano A. Marraffini

Version 0:

Reviewer comments:

Reviewer #1

(Remarks to the Author)

Hofmann et al. design and apply a high-throughput deep mutational scanning platform to determine impacts of all possible mutations in three components of a CRISPR/Cas enzyme coimply on nucleic acid capture and integration. They identify molecular constraints that conform with an Alphafold model of the complex and identify and validate mutations that enhance spacer integration and therefore antiphage immunity. Overall, I find the study to be well designed and performed, and appreciate the validation assays used to support the key conclusions from the mutational scan. I have only one minor comment:

1. Around line 204, is there a typo (perhaps the mention of Cas1 Y261R the second time on line 204 should be Cas2 R102F)? If not, could use a bit of editing for clarity of this section

Reviewer #2

(Remarks to the Author)

The manuscript by Hofmann et. al. reports a deep mutational scanning effort to improve the spacer acquisition efficiency of *S. pyogenes* Cas1-Cas2. The authors designed a sensitive selection assay to detect spacer acquisition and used it to scan through a deep mutational library, in which each residue in SpCas1, Cas2, and Csn2 has been flipped to the other 19 amino acid combination. The experimental setup was quite elegant and the authors made two rounds of effort to optimize the selection assay. Productive mutations were identified and validated in different settings. These include altering the host to evaluate the impact of the host factors; evaluating the improvement in anti-phage activity, and evaluating the fitness cost. The authors also made satisfying attempt to rationalize the mutational traits based on the existing structures and the alphafold models. This work is of general interest because 1) it serves as a good example showing how deep scanning mutagenesis can be utilized to improve molecular traits; and 2) it is a solid step forward in implementing Cas1-Cas2 based barcoding applications.

The authors did a good job in writing and figure preparation. It is a lot of data to sift through, however, in the end this reviewer found no major issues. There are some minor suggestions.

1. The authors rationalized the Cas1-R5 mutations immediately after result description, but they did not rationalize the beneficial mutations until the discussion. It is better to be consistent.

2. The rationalization of the M77H mutation was vague besides pointing out that a similar mutation also improved spacer acquisition for *E. coli* Cas1-Cas2. Does this mutation improve Cas1 dimerization, locking Cas1 dimer in the productive conformation, or improve Cas1-Cas2 affinity. It would be nice if the author could probe it further and get some Kd numbers. However, I will not make it a requirement because what's shown is sufficient for publication.

Version 1:

Reviewer comments:

Reviewer #2

(Remarks to the Author)

The authors have addressed all of my concerns. I recommend the publication of this study without any reservation.

The reviewer's critiques are in black, and our response is in blue. We thank the reviewers for their insightful comments and timely response.

Response to reviewer 1

Hofmann et al. design and apply a high-throughput deep mutational scanning platform to determine impacts of all possible mutations in three components of a CRISPR/Cas enzyme coimply on nucleic acid capture and integration. They identify molecular constraints that conform with an Alphafold model of the complex and identify and validate mutations that enhance spacer integration and therefore antiphage immunity. Overall, I find the study to be well designed and performed, and appreciate the validation assays used to support the key conclusions from the mutational scan.

We thank the reviewer for recognizing our work as "well designed and performed" and also for pointing out that our validation assays support the key conclusions. Here, we address the minor comment of the reviewer.

I have only one minor comment:

1. Around line 204, is there a typo (perhaps the mention of Cas1 Y261R the second time on line 204 should be Cas2 R102F)? If not, could use a bit of editing for clarity of this section

We apologize for the mistake and thank the reviewer for recognizing this error. We have corrected the manuscript accordingly and the section now reads as follows:

" To test the interaction between these residues we swapped them. The Cas1 Y261R mutation alone displayed a minor effect on spacer acquisition (Fig. 2e). In contrast, the **Cas2 R102Y** substitution generated a reduction in the incorporation of new spacers by approximately two orders of magnitude, which was partially rescued in the double mutant (Fig. 2e)."

Response to reviewer 2

The manuscript by Hofmann et. al. reports a deep mutational scanning effort to improve the spacer acquisition efficiency of *S. pyogenes* Cas1-Cas2. The authors designed a sensitive selection assay to detect spacer acquisition and used it to scan through a deep mutational library, in which each residue in SpCas1, Cas2, and Csn2 has been flipped to the other 19 amino acid combination. The experimental setup was quite elegant and the authors made two rounds of effort to optimize the selection assay. Productive mutations were identified and validated in different settings. These include altering the host to evaluate the impact of the host factors; evaluating the improvement in anti-phage activity, and evaluating the fitness cost. The authors also made satisfying attempt to rationalize the mutational traits based on the existing structures and the alphafold models. This work is of general interest because 1) it serves as a good example showing how deep scanning mutagenesis can be utilized to improve molecular traits; and 2) it is a solid step forward in implementing Cas1-Cas2 based barcoding applications.

The authors did a good job in writing and figure preparation. It is a lot of data to sift through, however, in the end this reviewer found no major issues.

We thank the reviewer for describing our experimental setup as "quite elegant" and recognizing the general interest in this work. We appreciate that the reviewer found that we did a "good job in writing and figure preparation" and that there were "no major issues". Here, we address the minor suggestions of the reviewer.

There are some minor suggestions.

1. The authors rationalized the Cas1-R5 mutations immediately after result description, but they did not rationalize the beneficial mutations until the discussion. It is better to be consistent.

We thank the reviewer for this recommendation. To be consistent, we have moved the rationalization of the Cas1 R5 mutations to the Discussion.

The edited Discussion section concerning this change now reads as follows:

"In contrast to the strong overall negative differential selection, we examined two regions at the Cas1-Cas2 interface that displayed relatively low amino acid preferences. One region corresponds to Cas2 loop 6, which, in the AlphaFold3 model, primarily interacts through the peptide backbone, forming a key interaction with the side chain of Cas1 R5. **Substitution of R5 to alanine completely abolished spacer acquisition. Given that residue changes at this position did not disrupt Cas1-Cas2 integrase assembly (Supplementary Fig. 12c), R5A may instead interfere with the assembly of the type II-A supercomplex or with some of its functions in spacer sampling and integration.**"

2. The rationalization of the M77H mutation was vague besides pointing out that a similar mutation also improved spacer acquisition for E. coli Cas1-Cas2. Does this mutation improve Cas1 dimerization, locking Cas1 dimer in the productive conformation, or improve Cas1-Cas2 affinity. It would be nice if the author could probe it further and get some Kd numbers. However, I will not make it a requirement because what's shown is sufficient for publication.

We thank the reviewer for this suggestion. While we did not further investigate the mechanism of M77H, we addressed a related question by analyzing the Cas2 Y5F variant, which was also enriched in our screen. We prioritized Y5F for follow-up studies because of (i) its proximity to the cation- π interaction and the backbone contacts of Cas1 R5 at the Cas1-Cas2 interface, both investigated in this manuscript, and (ii) its likely role in organizing the variable Cas2 loop, as suggested by structural models. We include the additional data in panel c of the updated Supplementary Fig.18 (previously called Extended Data Fig. 7):

The edited Discussion section with the additional rationalization of Cas2 Y5F now reads as follows:

"One of the most effective residue changes we identified was the relatively conservative Cas2 Y5F. Structural models of the Cas1-Cas2 integrase complex suggest that both Cas2 Y5 and Y5F contribute to the structural organization of the variable Cas2 loop through several van der Waals interactions – along with the analyzed cation- π interaction between Cas1 Y261 and Cas2 R102 and the backbone interactions of Cas1 R5 (Supplementary Fig. 18a, b). **Analytical SEC showed that both Cas2 Y5F and Y5A formed higher-molecular-weight complexes with Cas1 (Supplementary Fig. 18c), similar to the wild-type (Supplementary Fig. 12c), indicating that these residue changes do not affect integrase complex assembly. Instead, the enhanced activity of Cas2 Y5F may result from allosteric modulation of catalysis or structural effects at a different stage of spacer acquisition.**"

Reviewer #2 (Remarks to the Author):

The authors have addressed all of my concerns. I recommend the publication of this study without any reservation.

Thank you for reviewing our manuscript.